# Indigenous Meanings of Provenance in the Context of Alternative Food Movements and Supply-Chain Traceability: A Review

**Chetan Sharma** [1,*] , **Damir D. Torrico** [1] , **Lloyd Carpenter** [2] and **Roland Harrison** [1]

1   Department of Wine, Food and Molecular Biosciences, Lincoln University, Lincoln 7647, New Zealand;
    Damir.Torrico@lincoln.ac.nz (D.D.T.); Roland.Harrison@lincoln.ac.nz (R.H.)
2   Department of Tourism Sport & Society, Lincoln University, Lincoln 7647, New Zealand;
    Lloyd.Carpenter@lincoln.ac.nz
*   Correspondence: Chetan.Sharma@lincoln.ac.nz

**Abstract:** This article reviews the concept of *provenance* from both contemporary and traditional aspects. The incorporation of indigenous meanings and conceptualizations of *belonging* into provenance are explored. First, we consider how the gradual transformation of market*places* into market and consumer activism catalyzed the need for *provenance*. Guided by this, we discuss the meaning of provenance from an indigenous and non-indigenous rationale. Driven by the need for a qualitative understanding of food, the scholarship has utilized different epistemologies to demonstrate how authentic connections are cultivated and protected by animistic approaches. As a tool to mobilize *place*, we suggest that provenance should be embedded in the immediate local context. Historic place-based indigenous knowledge systems, values, and lifeways should be seen as a model for new projects. This review offers a comprehensive collection of research material with emphasis on a variety of fields including anthropology, economic geography, sociology, and biology, which clarifies the meaning of provenance in alternative food systems. It questions the current practices of spatial confinement by stakeholders and governments that are currently applied to the concepts of provenance in foods, and instead proposes a holistic approach to understand both indigenous and non-indigenous ideologies but with an emphasis on Maori culture and its perspectives.

**Keywords:** ethics; economic sociology; commodification; consumers; social movements; values

## 1. Introduction

There is a growing interest in food provenance amongst consumers for a variety of reasons, including self, economic, environmental, social and cultural well-being. Consumer well-being involves motives mostly related to food safety and health, such as preventing fraudulent practices, maintaining food safety throughout the supply chain, source, traceability, and reliable nutritional information. Economic well-being is progressively extending from simple profit and future financial security towards social spheres, where consumers want to know that if the trade was fair; farmers were not exploited; workers were healthy, safe and earning a living wage; and that local food sovereignty is not compromised. This extension morphs into environmental and social well-being that is defined by community appreciation of environment, farmers, production, locavore, social group, and/or indigenous knowledge, and increasingly incorporates a movement away from chemical-based, mass production systems. A glimpse of this can be noticed in the rising popularity of farmers' markets around the world, where the concept of food provenance is emphasized by the face-to-face interactions between producers and customers, which further helps to link locality with quality and ethics with the environment (Soon Jan and Wallace Carol 2018). Paralleling this rise in the popularity and number of local producer markets and interactions, provenance emerges as a tool to learn where and by whom food

was grown, gathered, or raised (origin), and how it was produced and transported, forming reassuring feedback that purchase decisions are both wise and just (CCEA 2020). Moreover, recognition of local farm ownership and information about moral economies are becoming important factors in consumers' purchasing decisions. The positive associative values of *the family*[1]-based structures (Andreini et al. 2020) in farm ownership compared to corporate farming entities also affect the perception of the *goodness* of food. For instance, it was found that the consumer population of Colorado associated a *good food* construct with short delivery chains and small-scale family-owned farms (Carolan 2020). Providing consumer information about the origin of food helps consumers reduce their anxiety about safety, quality, authenticity, ethics, and sustainability through perceptions and knowledge about the spatial, social and cultural characteristics of that provenance (from field notes 2021). For instance, information such as whether lettuce was field-cropped or hydroponically- or greenhouse-produced, fish was wild-caught or farm-raised, chickens were battery-housed or free-range, pigs were raised without the use of sow crates, beef was grass-fed or feedlot-penned helps consumers to make decisions about purchases based on their trust and belief in the quality and ethical nature of that production system.

According to the Cambridge English Dictionary, the definition of provenance is "the place of origin of something" (Cambridge University Press 2021). It is unsurprising, therefore, that most food provenance studies have been focused on the spatial meaning of origin and less on social wellbeing, connectedness, and cultural aspects of origin. This framing leads to an asymmetry of attention between consumer wellbeing and social welfare as well as a growing sense of disconnection with the land. While food provenance is often conflated with origin or place (Meah and Watson 2013), it has a broader meaning, especially if we consider it a response to modernity (Reid and Rout 2016). The meaning of provenance from the consumers' perspective has not been confined to the geographical aspect of origin only (Meah and Watson 2013), instead, it includes caring from local to distant farmers (Meah and Watson 2013; Soon Jan and Wallace Carol 2018); and from human to non-human actors (Jiang et al. 2020; Meah and Watson 2013). Provenance has a flexible relationship with the actual conditions of production (Smith Maguire 2013) and it should not be confined to an extrinsic attribute; instead, it is a negotiable, accomplished, integral quality that is socially constructed (Beverland and Farrelly 2010; Grayson and Martinec 2004), and includes spatial, social and cultural dimensions (Morgan et al. 2008). The entwined and interdependent relationship of nature and culture can be better understood in relation to *animism* and cultivated by employing animist metaphors. Unfortunately, earlier researchers linked animism to "primitive" or "tribal" constructs and even considered studies of ontology to be unscientific, resulting in the jettison of the *animism* concept, despite its presence in many cultures (VanPool and Newsome 2012). However, from the vantage of connecting social and cultural dimensions of place to provenance, which is seen as a means of revealing and restoring relationships with the wider world, animism can provide critical contributions by providing a mixture of intrinsic and instrumental motivations (Rout and Reid 2020). The holistic significance that consumers attach to food provenance can impact the economic development of local or corporate production companies, farming systems, transport systems, social relations, and has social welfare consequences on local producers, farmer-owners, and their workers (Soon Jan and Wallace Carol 2018). Nevertheless, the concept of provenance can be considered and delivered in isolation, but the consideration of other beyond spatial spheres would provide a more qualitative understanding of food systems, their involved stakeholders, and perhaps the human–nature relationship. This relational understanding can be authentically realized and cultivated through indigenous animist approaches. Ongoing, animism seeks to connect but its roots resting in modernism, which manifests through conceptual fragmentation, defeating its original purpose. Hence, although the indigenous world celebrated, simultaneously it also suffers from a level of othering in terms of comparative analysis as far as food provenance goes. Our consideration of this topic engages with the otherness of indigeneity to explore and explain new paradigms and to suggest a new potential future where bicultural (settler or colonial hegemonies and

indigenous peoples) manifestations of provenance posit significant advantages accruing from such an approach. In this context, this review focuses on the indigenous meanings of provenance manifested through animistic approaches in place-based epistemologies, and its role in establishing moral economic practices by making authentic connections. Finally, a Māori perspective on production and entrepreneurship systems will be presented as an applied case of indigenous provenance concepts on the understanding of food systems.

## 2. Transformation of Market*places* and Consumer Activism

Interestingly, both big producers/suppliers (represented by mass-production companies) and consumers are addressing these concerns differently, through embedding geographical and socio-cultural dimensions into their considerations. In this respect, big producers increasingly emphasize quality in *defensive localism* (certifications) aspects, supported by *environmental-corporate food regimes* and *certifications*, which were argued by Watts et al. (2005) as *weaker alternatives* compared to *alternative supply chains*. Consumers, on the other hand, are increasingly opting for quality in *alternative supply chains* (farmers market) and *alternative economic practices*, respectively (Rosol 2020; Watts et al. 2005). Thereby, unconsciously, both big producers and consumers are involved in *value* production (Page 2017), which further cultivates the opportunity for the economists to expand the consumer world. Ongoing, this expanding consumer world, manifested through consumerism, seeks to make you believe that there is enough availability of inputs (seeds, water, and land) and markets can be transformed without encouraging agency, anti-consumption, and an ascetic lifestyle.

The major changes in the nature of labor, land, and animal use that transpired in the post-war mid-20th century era raised moral concerns, which increasingly became an issue of significance. For instance, husbandry once meant providing animals with optimal conditions dictated by their biological needs, but as the nature of agriculture deviated to a more industrial enterprise animal welfare became a subsidiary issue (Rollin 2007). Nevertheless, it may be difficult to pinpoint the emergence of ethics, specifically of production or related to animal welfare in general; however, it does not mean that it emerged *ex nihilo*. A plausible and apparent reason could be the customs or rituals, practiced in the form of customary standards of right or wrong conduct (in the form of totemism), developed to ontological, social, economic, and spiritual exchanges, and which subsequently laid the foundation of ethics. These succeeding ethics developed for self-interest were later extended towards animals as "subjects with rights" in the later 20th century, through animal welfare activists and multispecies ethnographers via discursive strategies of trans-species recognition, to deal with the new upcoming issues of capitalization or factory farming (Ogden et al. 2013; Rollin 2004; Rollin 2007). Innate anthropomorphic tendencies of humans for ethics transfer may also be a contributing factor for animal welfare-related ethics. Informed by animal biology and behavior, Rollin (2007) proposed an ethical framework of *natural laws of behavior* or *telos* apropos of subject(s) in question, such as "fish got to swim, birds got to fly." This said the *telos* concerning human nature and their multispecies encounters infused deep cultural values often highlighting the epistemic tensions between humans and nonhumans. Undoubtedly, social media played a significant role in the information flow, meaning the development and strengthening of ethics and ethical consumption in a transnational context. In the ongoing discourse, the engagement with consumption ethics can be seen as an outcome of the dialectical tension between globalization, big corporates, long chains of mass production and localization, small farmers, ethical and sustainable production (Campbell 2009; McMichael 2009).

Concomitantly, *neoclassical* economists saw this progressive loss of connection as an opportunity to simply see self-interested behavior and build "institutionally separate" economies[2], promoting the organization of production by the *market*[3], which has institutional foundations in commodification. Fueled by a lack of knowledge of production systems (Brombin et al. 2019), the consequent permissive environment helped large-scale industrialization and globalization of food supply chains to emerge. This lengthening

of mainstream supply chains inevitably created significant issues or externalities[4] in the market highlighted by food scandals (Barbarossa et al. 2016; Liu and Ma 2016; Roy et al. 2018); loss of trust (Watts et al. 2005); worsening conditions for small farmers (Allaben 2020); unethical practices (Rollin 2004; Rollin 2007); or other individual reasons, such as health issues. Such an economic system assumes land, labor, and money as "elements of industry" and are subject to market mechanisms of production, sale, price signals, supply and demand, and exchange through buyer and seller (Gemici 2007). This separation of the *market* as a principle from the market as *a place* motivated by factors of "purely economic" or "market forces" in nature (Bestor 2004), which broke the thread linking consumers and producers and created opportunities for fetishization. The original meaning of the word *market* was a market*place*, described by both a geographic place and a localized set of social institutions. However, in the age of globalization, international freight networks and increased use of "borderless trade relationships" created by networks of countries, the market has expanded to mean anything that affects the price, demand or supply of a good or service.

The followed expansion of the *self-regulated market* to accommodate labor, land, and money as commodities spurred self-protection in society (Gemici 2007), as the two cannot be separated either conceptually or empirically (Busse and Sharp 2019). In reaction, a protective response is observed in the form of "shopping and supporting local" through food miles, farmers-market, low footprints, or alternatives, which may include diet and lifestyle change or a decision to fully engage with provenance features of foodstuffs.

These emerging movements are unfolding new patterns that shift away from focusing on the material exchange between producers and consumers (Schermer 2015). A myopic view equating market exchange to personal material wealth is problematic as it overlooks opportunity costs, negative externalities, and unintended consequences (Laczniak 2017) of those market exchanges. Leading economists echo the need to make hidden costs embedded or implicit in macro-marketing systems or transparent sub-systems so that the full complexity and heterogeneity may become explicit instead and thus can be better understood (Laczniak 2017; Lusch 2017; Varadarajan 2020). In an insightful commentary, the late Robert F Lusch posited that the essential nature of *Homo sapiens* is to exchange (Lusch 2017), and as such, they engage in exchange with each other. By extension, the social function of memory becomes to exchange memories or information as a means of creating social bonds and thus creating tools to facilitate exchange, such as language, facial expressions, or "technology" in this context and form institutions to coordinate the exchange, such as marriage, school, and social groupings and/or social media. All of the costs in these exchanges are unseen, unrecognized, or unpublicized, which become evident later in the form of externalities and unintended consequences.

### 2.1. Meaning of Provenance in Indigenous Rational

The indigenous perspective of food provenance offers important insights about its multi-dimensional woven universe, which has never fragmented its long traditions with the "living web of the world" (Reid and Rout 2016; Spiller et al. 2011). The *connectedness* and *belongingness* cores of indigenous perspective contradict the dominant contemporary/mechanistic ideology, which advocates that the purpose of businesses is merely to produce material wealth, with economic and environmental externalities largely ignored. This incomprehension invokes Karl Polanyi's vision of *embeddedness* that all economic systems are embedded and enmeshed in social relations and institutions (Dewey 1958), which was later advanced by his followers under the concept of *substantivism* (Gemici 2007). Informed by the primary driver of the *embeddedness* concept and thus rejecting the alleged demarcation between economic and social phenomena, this methodological principle is taken to discern the changing place of the economy in alternative food movements. Food is conceptually more than a commodity, *de facto*, it can be considered as a statement of place or culture (Stevens 2020); in Polanyi's terms, labor is "human activity in life", the land is "nature", money is a "token of purchasing power," (Polanyi and MacIver 1944), and

therefore constricting the essence of food to "production for sale" is problematic. Moreover, food is sold as a part of an experience (Aiello and Gendelman 2008), or a lifestyle (Giorda 2018) where a passion or family tradition and collectivist values (Tretiakov et al. 2020) tend to override other desires such as those dictated by vertical hierarchy. An act motivated by such transcendent values instead of the simple utilitarian provides context to some specific kind of economic activities[5], yet still including the notion of *embeddedness*. What makes a value transcend is related to the vision of a better world, whether envisaging the future or remembering the past. For instance, the provisioning of food by *back-to-the-landers* and *freegans* via tapping into pre-capitalist subsistence patterns is a constructed vision that is shared by those using this term (Gross 2009). The physical abstraction of the consumers from their foods led by globalization and industrialization has disrupted the interaction between humans and nature (Reid and Rout 2016). The loss of connection frees the consumer from the guilt of association with deforestation, loss of indigenous lands and culture, and any other environmental degradation, but it removes the producer from accountability for the same.

Considering the *self-regulated formalist* economy as an outcome of a society whose modernist construct opposes the *relational* understanding, Reid and Rout (2016) argued that provenance would be likely to remain a marketing tool in such a society until we understood and conceptualize provenance using indigenous cosmology. Building on this argument, a concept of *animism* is explored that counters the abstraction by bringing a relational understanding of the world and can empower provenance. In the process, recent scholars have been expanding the theoretical and methodological implications of *animism* and other ontologies (Harvey 2005; VanPool and Newsome 2012) into art and performance (Braddock 2017; Porr and Bell 2012), psychology (Mays et al. 2020), religion (Laack 2020), cognitive science (Núñez and Cornejo 2012), food sovereignty (Ritchie 2016), biodiversity and in situ conservation of genetic resources (Gonzales 2000). The fundamental principle of *animism*, especially new *animism* is that "life is always lived in a relationship with others", and this notion refers to a concern of knowing how to behave appropriately towards different natural entities, where some of them are humans (Harvey 2005). Expectations about the behavior appropriate to those in a relationship of a particular form may vary culturally and it is explicit in Thompson's invocations of "what *ought* to be men's reciprocal duties', and these are likely to affect their dealings with each other ( Carrier 2018; Thompson 1971). The *relational well-being* metaphor of indigenous thinking, for instance, the Māori values and activities in which air, land, water, or fire are culturally and spiritually connected (Stein et al. 2018), helps bridge this Cartesian division (Sillar 2009) and have been the recent focus of the *Te Taiao*/Environment movement in Aotearoa New Zealand (ANZ). Since its official launch in mid-2020, *Te Taiao* has been embraced by an increasing variety of ANZ food producers, led by the Ministry for Primary Industries and as a producer-led, government-supported initiative, represents a positive, multicultural, inclusive environmental movement (MPI 2021).

It has been long demonstrated by archaeologists that the boundaries between human and non-human animals are much blurrier than our Western contemporary dichotomies concepts (Ingold 1994; Ogden et al. 2013). In contrast to the anthropocentric notion of western cosmology, the *Quechua* and *Aymara* of the Andes visualize mountains, rivers, waters, women, and men as alive and incomplete (Gonzales 2000). This construal of *incomplete* allows dialogue, reciprocity (Gonzales 2000) and can be understood through the etymology of *Yanantin* (dualism), "for your conjugal pair", by the twice action of spilling a few drops on the ground with the name of the receiving divinity (*ch'alla*, libation), considering the highest peak as male and the second highest peak as his female counterpart. Together they constitute the *Achacilas* by offering the two handfuls of coca leaves into two cupped hands in a ritual (Platt 1986). The term *Yanantin* is made up of the stem *yana-* which means "help", while its suffix *-ntin* means "inclusive in nature, with implications of totality, spatial inclusion of one thing in another or identification of two members of the same category" (Platt 1986). *Aymara* beliefs are manifested in the sacredness of nature,

which is expressed through *animism* in the form of rivers, mountains, and animals (Núñez and Cornejo 2012). The careful work of indigenous communities to build up their food, educational and economic systems while maintaining their cultural practices is often undermined by external forces (Turner et al. 2013) invoking a transcendental dimension that inherently separates people and physical entities from the environment in which they exist (Núñez and Cornejo 2012). Likewise, this is expressed through embracing the cultural impetus to *kaitiakitanga* (environmental stewardship/guardianship) and the animist concept of a pervasive *mauri* (Mead 2016) in the Māori world.

A recent modern movement that houses a similar underlying indigenous meaning of worldview is *deep ecology*, which challenges humans' superiority over nature and favors the decentralization of humans in ethics and theory via *posthumanism* (Erdős 2019; Ogden et al. 2013). *Posthumanism* offers a new epistemology that is not anthropocentric and does not pose humans as coherent, singular, and external to beings considered "of nature", such as other animals and from "naturalized humans," such as indigenous peoples (Castree and Nash 2006; Ogden et al. 2013). This *deep ecology* concept includes a holistic principle, which states that relations between entities are more important than the entities themselves, and reality should be conceived as an intricate web where our *self* should include all other living beings. Similar conception can be found in many cultures, such as the *Garo* tribal people of India worship of the sun and moon, which is characterized by their *naturalism* ( Sharma 2004), *Aymara* of Andes perform rituals to *Pachamama* through offerings involving special arrangements of coca leaves and maize and likewise for the Māori of ANZ, through the use of *mauri* stones (Best 1982), *whakapakoko rākau* (god sticks) (Skinner 1922) and first yield crop sacrifices to the *atua* (deities).

The tribal belief of soul (*animism*) and impersonal supernatural powers (*animatism*) make native tribes of India aware of the concepts related to exploitation, remorse abnegation, expiation, giving-of-self, and communion. These enable them to live life in a synchronous moral order established at the beginning of human history. For instance, the *Korawa* tribe practices the offering of sacrifices to please divine entities in exchange for having a productive year of harvesting. Similarly, the *Kutia Khand* or *Khond* tribe of Odisha, India practice sacrifice towards mother earth and this is a common phenomenon among other Indian tribes (Sharma 2004), which teaches them about the reciprocal flow of life and supernatural powers. These sacrifices (may have been of an animal, plant, or human) ranged from emphasizing divine favors, minimizing nature's hostility, and promoting the earth's fertility (Faherty 2020). The tribal beliefs of embodying impersonal supernatural powers in bones, feathers, stones, and bead stones are characterized by *animatism*, making indigenous people connect with their local environmental niches. Customs, as vernacular beliefs relate to the ecological processes involved in the activities of the *Aymara* community are strongly tied to the productive cycle of *Pachamama*—Mother Earth and *Pachamama* not only feeds but sustains life, and humans are part of herself (Núñez and Cornejo 2012).

The customization of the totem is an essential characteristic of most tribes, and forms a baseline for many great extensions of human–nature relations (Wagner 2018), but for this purpose, *Garo*[6] will be taken as an example (Hossain 2019), and totemism is seen as a metaphor of inter-connected axiology's, unity, cooperation, and emotion, where tribes share an emotional bond between flora and fauna of their environmental niche. These metaphors foster the consciousness of unity among the members of the tribe and nature (Sharma 2004). The cultural construction of kinship among aboriginals (Dudgeon and Bray 2019) extends to help the conservation of places, plants, and animals by forbidding exploitation and consumption of certain components of the environment (Eneji et al. 2019). Assignment of forbidden places, such as sacred groves in Mangroves, India, evil forests, or ponds helps protect biodiversity (Mitra 2020) and give the opportunity to nature for repair. Another instance of totemism is where certain trees are treated as totems and were never felled, due to beliefs that such trees were associated with water sources, having medicinal properties, associated with bad omens, ancestors, or were positively associated with luck and wealth (Ayaa and Waswa 2016; Mitra 2020). In identifying a sacred animal or plant,

the anthropic promote communion by establishing taboos and so cultural rules to bring an internal structure and rule to living and harvesting.

In Arne Naess' view, founder of *deep ecology* philosophy, there are no differences between humans and other living organisms, and our *self* is virtually identical to others in nature (Erdős 2019). In the same vein, several related endeavors such as *object-oriented ontologies*, *hybrid geographies*, and *post-structuralism* have been engaged under *multispecies ethnography* to reconsider nature and society and favor experimentation with alternative epistemologies such as *affect* and *non-representational theory* (Ogden et al. 2013). Reid and Rout (2016) echoed the role of animism in providing integrity to food provenance and without which there would be a danger of generating false realities through commodity fetishism. This said the modern concept of provenance is fundamentally groundless, where it is being considered as just another measure or tool to make profits in food businesses.

### 2.2. Meaning of Provenance in Non-Indigenous Rational

Since the early days, (around the 18th century) of westernization[7], metaphors of progress for western civilizations were related to flat-landing, sighting, rational deduction, linear ordering, universalism, dualism (Cartesian), and capitalism (Fox 2006; Johnson and Murton 2007; Rout and Reid 2020; Spiller et al. 2011), which, in the empirical world, was driven by self-interest, productivity, efficiency, knowledge, medical progress and product safety (Rollin 2007). Taking these metaphors in a social and conceptual framework serves to distance consumers from nature, culture (Johnson and Murton 2007), and taste ( Beans 2017). This remoteness could be multidimensional, including geographical, social, organizational, or institutional factors (Dubois 2018). This type of framework, irrespective of the social, economic, or geographic background, omits all holistic provenance views and displaces plants, animals, or people from their landscapes (Johnson and Murton 2007). The non-indigenous meaning of provenance favors reductionism and the simplifying strategies of this approach meshed well with the ecological and social simplicity of standardized provenance systems in terms of objective biological facts of spatial or geographical identity. Several examples of this type of pairing can be found in the related contexts, such as mono-cropping, the green revolution, individual health and stress management, and others. (Gonzales 2000; Sherman 2020). A fragmented visualization of *well-being* in non-indigenous notions focuses only on the "wellness" half of *well-being* via practicing individual biological facts but does not address the root causes (Sherman 2020). Likewise, the machine metaphor of non-indigenous notions limits cognition regarding "sustainability" because its modeling frames nature as predictable and controllable (Rout and Reid 2020) as well as locating place as out-of-sight and beyond comprehension. The quick fixes or alternative measures and short-term improvements of the *shallow ecology*, a modern parallel of the non-indigenous philosophy (Erdős 2019), apply technological solutions to the environmental problems without questioning the place of *self* in nature. By following the meaning of the contemporary framework, provenance without landscape, culture, and nature can eventually be sized up as a mere cynical marketing buzzword. In this contemporary approach, provenance is identified as a piece of information or as a positioning device for differentiating brands across a range of markets (Smith Maguire 2013). Such an approach of commoditizing ethical values (Goodman et al. 2012) by modern marketers can severe the genuine spatial, social and cultural connections in food systems ( Reid and Rout 2016). A significant emphasis on the ancillary services of provenance, such as safety, transparency, geotagging techniques, data collection methods, elemental and molecular profiling, and DNA fingerprinting is shifting away the attention from its core values of *connectedness*. Indeed, the scope of provenance is being limited to geographical indications and traceability (Soon Jan and Wallace Carol 2018).

An operational definition of provenance, employed by modern entities, tends to treat provenance as an objective quantitative fact by providing information such as the country of origin (COO), greenhouse gas or water use footprints, agrobiodiversity, and in situ conservation indexes. In the same vein, traceability has been introduced as a management tool

to underpin the transfer of accurate records from the farms to the plate of end-consumers (Barling et al. 2009). The use of explicit governance and a varied range of certifications such as the use of *Denominazione di Origine Controllate* (Italy)/*Appellation d'Origine Contrôlée (France) for wine*, and ethical and environmental accreditations (including certificates of Organic farming practice, Fairtrade, and biodynamic methodologies), and third party certifications to establish authenticity represent this contemporary approach (Giorda 2018). However, these approaches are developed under transparency and traceability measures, and the credibility of these systems has been challenged globally due to recent high-profile food scandals (Barbarossa et al. 2016; Liu and Ma 2016; Roy et al. 2018). Watts et al. (2005) revealed that the existence of a certificate *de facto* demonstrates a *disembedded* production[8] (Page 30 of reference), but this revelation simultaneously fails to acknowledge the fact that all supply chains are embedded to some degree in specific territories and particular social contexts (Bowen 2011). The fetishization of land or place or means of production is against the traditional indigenous meaning and is covered extensively in the cooperative *alternative* food movements (Lee 2000; Rosol 2020; Watts et al. 2005). Commerce, just like certifications, reflects a purely economic relationship that dissolves as soon as the transaction is completed (Carrier 2018). However, in a moral economy, at least in principle, the relationship between parties interacting for simple economic and utilitarian reasons is not reducible to what is transacted; instead, it encourages consumers and producers to look at transactions in terms of relationships and their histories. Pratt (2007) emphasized that authenticity is a feature of the rooted and ancient, not of the modern and culture as this concept does not have a monetary connotation (Giorda 2018; Pratt 2007). This implies that such activity is motivated to a significant degree by the relationship in which it occurs. Simultaneously, authenticity is not an attribute inherent in an object, but it is better understood as an assessment made by a particular evaluator for a specific context (Grayson and Martinec 2004), and it can be only achieved with indigenous knowledge, often relying on immediate and remote family connections or tribal connections (Tretiakov et al. 2020). Certifications and scientific formulations are modern in character (Busch 2000) and are less appealing alternative options (Sage 2003), considering their vulnerabilities to subordinate within the traditional food supply chains. Conceptual metaphors of these problem-centered modern derivations rest in empiricism, which is practiced extensively in the classroom.

The contribution of the *Experimentalism* and *Progressivism* schools of thought into the aforementioned non-indigenous concepts should not be overlooked, as *experimentalism* education emphasizes inquiry, consensus, and process rather than authentic freedom (Lieberman 1985). The *Experimentalist* scientifically tests solutions to environmental problems and is primarily concerned with gathering factual evidence, while *Existentialism* tries to find answers by responding emotionally to the environment. The problem-centered notion of *Experimentalism* and *Progressivism* represent a detached way of dealing with human experiences, where a "low-conflict" rapport with others (in society) is the governing principle. The unfolding of the personality, including both rationally and emotionally, in the *Existentialist* school is critical but *Enlightenment theory* denies this totality and instead led to the optimistic belief that rationality alone, with the accompanying belief in science, experimental methods, and progress, would perfect life and create human happiness (Sherman 2020). In the 18th century, the thinkers of the Scottish *Enlightenment* invoked the idea of a *formalist* economy by arguing that people's activities in the economic realm, especially their dealings with others, should not be tainted by sentiment (Carrier 2018). This notion of the self-regulating and *dis-embedded* nature of market exchange and market economy, so-called *neoclassical economics*, reflects Polanyi's *embeddedness* concept (Gemici 2007). The merchants of the 18th century created the factory system through the introduction of "elaborate specific machinery and plant" (Polanyi and MacIver 1944), which intensified specialization and a continuous supply of production factors (Gemici 2007). Complementing the demand of big business and mass production, *Experimentalism* and *Progressivism* emphasized the need for quantitative, fragmented measures to fit precisely into the scheme of the organization.

The concept of provenance is much broader than merely the geographic domain of place, and thus, confining it into one dimension alone is nothing more than pandering to the belief system of *neoclassical* economists. This said, provenance can be explained without reference to the values, simply based on objective biological facts, led by a *formalist* economy, but this would create perturbations among producers, mediators, consumers and would be eventually lost to capitalism and commercialization.

## 3. Provenance and Its Interaction with Place and Terroir

### 3.1. Place and Provenance

The notion of place is to *bring together* but the operating principles resting in traditional food supply chains face the paradox of place-disruption (Busse and Sharp 2019) since foundations of the traditional food supply chain are based on "thinning out the conceptualization of place" (Feagan 2007). Attention to spatial practices is not new; *de facto*, they precede the now conventional, globalized food supply chains. Increasing interest in food re-localization follows a prolonged period of what might be termed "de-localization" but is commonly referred to as *Progressivism* or *Productivism*. However, recent growing interest in food re-localization ties in with increasing global economic and political instability, environmental degradation, climate change, and levels of social inequality (Rosol 2020; Watts et al. 2005). The corporatization of agriculture and commodification of nature diminished the physical interaction of consumers with the place and practices, resulting in disconnection in the *value* of what people grow and eat (Stevens 2020).

Given the problematic socioeconomic impacts of disconnection, the agrifood sector is testing new ways of engaging the *place* in businesses. In the ongoing state of affairs, food provenance is increasingly offered as a solution despite evidence that consumers regard labels and certifications with ambivalence, if not outright suspicion (Eden et al. 2008; Smith Maguire 2013; Watts et al. 2005). Food provenance is very much driven by geographical indications (GIs) and traceability measures, supported by PDO and PGI (Soon Jan and Wallace Carol 2018). Simultaneously, however, it seems that consumers who are buying local are not actually buying local in the detached objective biological fact form offered by *conventional* markets or sometimes by *alternatives*, instead they are opting for communion. The wistful longing of self-realization (coming into *being*) and *rematriation* allude to different concepts, such as local, authentic, ethical, or *humane*. However, these concepts are hard to foreground in Western institutions, and they often are subsumed in some other contexts that are not under the Western measure of ethics. Examples of this cosmovision difference can be observed in many instances; however, those of *biodiversity* and *conservation* especially merit discussion here. Under Western cosmology, *biodiversity* has its origins in the field of conservation biology, which refers to the variability among living organisms and the ecological complexes of which they are part, whereas under non-Western cosmology *biodiversity* is best understood as "Mother Nature" and has obviously broader measures that never concerns a Western notion of *biodiversity* measure. Indeed, the Western concept of "conservation" cannot be translated, yet indigenous equivalents are often framed under the terms of "respecting nature" or "taking care of things", but whether they fall under the Western measure of *conservation* is a long-standing question (Gonzales 2000). Ecologists and economists coined the concept of ecosystem services to make biodiversity conservation intelligible to decision-makers versed in economic thinking but unlike indigenous cosmology, ecosystem service grounds environmental decisions in financial and economic modes of reasoning which further delimits ecosystem to economic benefits (Stevenson et al. 2021). In indigenous cosmology, "Mother Nature", "respecting nature or taking care of things" is not a discrete or special activity; instead, they are all interconnected and that is why they are not apparent as such or measured specifically. For example, agrobiodiversity, which is a specific contemporary policy, is nurtured under the culture of *native seed* in the indigenous world (Gonzales 2000). The Māori worldview considers all living and non-living in the ecosystem to be interconnected and this relationship between ecosystem health and peoples' wellbeing is revealed in the

Ngāti Whatua tribal whakatauki/saying: *Ko ahau te taiao, ko te taiao, ko ahau*—I am the environment and the environment is me (Rangiwai 2018). People, plants, and animals are all descendants of *Ranginui* (the sky father) and *Papatuanuku* (the earth mother) and their children, which means humans are, therefore, intrinsically linked with biodiversity. At a metaphysical level, this example helps understand the fundamental disjuncture between modernism and animism that manifest on the ground by the clash between the aims of the global food industry and the desires of consumers.

*In-place*[9] human *values*, experienced by consumers in the context of *alternative*, roots production in particular places. Hence, provenance, which is often manifested through GIs, should be fundamentally tied to the notion of *place*. In order to preserve the link to place, provenance employs a specific set of rules, manifested through regulations and certifications, to protect the in-place values developed over the course of time. However, simultaneously the danger of abstraction of some of those moral experiences of producers by the currently dominant conventional market system is massive[10]. In the realm of abstraction, these in-place human values transform to in-place *beautifications*, which are hand-picked, regulated, and articulated mischievously by those involved in the current hegemonic market system. Advancing this, McEwan et al. (2017) called for the deeper understanding of producers' moral experiences that are strongly embedded in the local social and cultural relations for the effectiveness of certifications in improving the livelihoods of farmers. These moral experiences, which are thoroughly embedded in local social worlds and cultures, foster *reciprocity*, a characteristic feature of local food systems (Reuter 2019), and prohibit utilitarian approaches to reduce or confine exchange to commodity transactions only (Orzes 2017). Reciprocity, as an organizing principle of *social economy*, manifesting through the realm of producers (Moberg 2014) mitigates economic risks based on an ethos of mutual care, justice, cooperation, solidarity, and trust (Kumbamu 2018; Reuter 2019). This non-universal or varied notion of mutual care in the realm of working conditions resists standardized monitoring mechanisms (Moberg 2014; Orzes 2017), which are otherwise developed and cultivated by auditing firms. Busse and Sharp (2019) made those shared values explicit and visible through *morality* in the context of transactions, what Carrier (2018) previously calls "moral economy", occurring in the Papua New Guinea marketplaces. By foregrounding the struggle of Caribbean banana farmers to persist in agriculture, what Moberg (2014) evoked is the paradox of *economic morality* where Fair Trade certification frequently violates farmers' moral economic relationships that occur through the realm of consumer choice (Flachs 2017).

In-place, *values* might have an emotional nature, such as the desire for warmth and nostalgic motivations (Sichtmann et al. 2019) or other influences which are more socioeconomic in nature, such as uneven economic and geographic opportunities, cultural ruptures, favoritism, and others. (McEwan et al. 2017). Trust has been observed as the main driver of alternative food networks (Delicato et al. 2019; O'Kane 2016), manifested through continuous embedding practices in the realm of local and by certifications in the realm of distant markets (Bowen 2011; Flachs 2017; Giorda 2018), whereas convenience, price, and value for money were cited as drivers of the mainstream food systems (Delicato et al. 2019). Continuous embedding practices, manifesting through place-specific moraleconomic complexities, reinforce the feeling of community among actors and enable the notion of preservation using a distinct sense of place (Boltanski and Thévenot 2006; Giorda 2018). Through horizontal linkages, prevalent in short food supply chains (SFSCs), these embedding practices foster trust-based relations between actors, and thus, they ensure more value of the product to the producer and a closer connection between producers, processors, and consumers (Bowen 2011; Delicato et al. 2019). SFSCs rely on the interaction, communication, and connection between producers and consumers; thereby, reinforcing the notions of social capital (Kneafsey et al. 2013). This increased interaction (RucabadoPalomar and Cuéllar-Padilla 2020) gradually approaches a stage of mutual respect and trust, which is not apparent in contemporary extended chains assuring trust through regulations (Flachs 2017). An embedded food system provides opportunities to reduce the

distance between abstract ethics, regulated by certifications, and the moral experiences of producers to help consumers engage in ethical and sustainable food practices (O'Kane 2016). In return, this favors decentralization and shared values. Since in-place values are unique to the territory[11], referring to provenance rather than place would be of no significance to the producers and indeed regulators.

By connecting the consumers back to a *place*, provenance is addressing the anxiety that is experienced by many Western consumers (Campbell 2009; Taylor 1991). Provenance has been framed previously as a tool to mobilize local food and contemporary gastronomic trends (Gyimóthy 2017). Having laid out a general schema of provenance in the *place*, the disjuncture between place and provenance is creating a potential breach in the discourse of moral economy. Provenance, therefore, not by itself but in a connection with place, emerges as a way of revealing and restoring the relationships with nature for those estranged by modernity, as it helps turn food from "nowhere" into "food from somewhere" and in the case of the Countdown supermarket chain, from "someone" who might be their neighbor or friend, but is noticeably and demonstrably *local* (McMichael 2009).

*3.2. Terroir and Provenance*

*Terroir* is a ubiquitous term in the wine world, as almost all English dictionaries have *terroir* definitions centered around wines such as, the Cambridge University Press, which has the meaning of *terroir* as "*the special character that a wine is thought to get from the particular place where the grapes were grown to make it*". The etymology of the word *terroir* is French and means "soil" or "land", but when this word connects with winemaking, the meaning increases in complexity. For wine-growing, *terroir* is not confined to spatial limits alone but covers aspects of the *place*. The complex social and cultural dimensions of *terroir* can be illustrated in Vaudour's (2002) four different types of *terroir*: (1) nutriment (a vertical relationship between soil, plant, and atmosphere), (2) space (territory, appellation, and historical geography), (3) slogan (images of country life) and (4) conscience (qualities of country identity, ancestry, heritage, and tradition) (Vaudour 2002; Warman and Lewis 2019). This word was first employed in 1863 in reference to wine as *goût de terroir*, a phrase that translates to "the taste of the soil" (Matthews 2016). The rich and relatable nature of the word *terroir* makes it difficult to "freeze" its meaning in one well-fitted general category.

*Terroir* has been used as an instrument for identifying the qualities of wines in terms of geo-climatic origin and authenticated methods of production. The notion of *terroir* is not merely about a quality index, but it has a symbolic capacity to articulate place identity by knitting the social, cultural, and natural characteristics of a region with gastronomy. However, this notion of *terroir* does not reconcile well with the ongoing corporatized and mechanized new world of winemaking (White et al. 2009). One apparent reason for this paradigm shift could be the change in the nature of the economy from market*places* to markets[12]. Coordinating with globalized change, new emerging markets and environmental niches are pushing the traditional systems, and many old geographical identities (GI) are losing their significance while new emerging GI's are gaining importance, such as the emergent centers in Oregon, the USA, and Central Otago, New Zealand (Carpenter 2016). Places with no heritage can be promoted strategically for the *terroir* construction based on its distinctive spatial specialties, such as agrarian, gastronomy, flora and fauna, social, culture, and heritage (Gyimóthy 2017) and this differentiation based on new *terroir* can help to preserve the genetic diversity of different plants, foods, processes, cultures, places, and identities.

Similar to place, *terroir* is an ambiguous and multifaceted concept, articulating the uniqueness of local produce by imbuing them with exclusive, quality-warranting connotations and properties (Bowen 2010; Gyimóthy 2017). *Terroir* is a powerful controller of the perceived enjoyment of tradition, history, memory, people, and product, which can be independent of the intrinsic sensory characteristics of the product. Terroir is the guidance of the place on the product, both biophysical and socio-cultural characteristics, and its resulting sensory distinctiveness, whereas provenance conforms to authenticity (Warman and Lewis

2019). Provenance serves as an intermediary to experience *terroir* and both of them need to work in tandem for creating the wine-place experience. Bowen (2010) wrote that "(sic) GIs are fundamentally tied to the notion of *terroir*. GIs are a potential tool used by local actors to counter negative effects of globalization". Consumers who learned to rely on provenance to predict wine quality are willing to pay a premium price for these products. Globalization made *terroir* typicality accreditation increasingly popular and provenance offers unique artisanal advantages for agricultural products against mass-produced industrial foods.

Since 2011 several New Zealand wine producers have chosen to embrace the Māori concept of *tūrangawaewae* (lit. "a place to stand"; the Māori conceptualization of belonging to place) instead of *terroir*, simultaneously adding location, place, "New Zealandness" and indigenous expression and interpretation to the buyer (Brown 2017). Generally, policies administered under western cosmology tends to "environmentalize" issues and tends to put emphasis on the patches of nature (such as in situ conservation, see Section 3.1) and short-term economic gains rather than focusing on the population outside or within targeted regions, which coevolve with these environments. Informed by the fact that the indigenous world has many strategies and that they are embedded within the immediate cosmology, both *terroir* and *tūrangawaewae* can be seen as epistemologies comprehending or appreciating the immediate nature (Figure 1). Similar epistemology can be found among Incas, who made observations relevant to the agricultural cycle within the local geographic context (Strong 2012). Likewise, Māori's cognitive template, *whakapapa* includes a folk taxonomy of the culturally important biota in a particular place (Roberts 2012). Since *being* Māori is different from the cultural construct that is French, the mere replacement of *terroir* with *tūrangawaewae* as a marketing tool poses ontological problems. The question of *being* in an ontological sense is critical here and epistemology must be in the service of ontology (Dall'Alba and Barnacle 2007; Sherman 2020). How to realize *being* is a separate question, which will be out of the scope of this work, but readers are highly encouraged to look for existentialist schools of thought on this topic and the related role of higher education (Lieberman 1985; Sherman 2020). Additionally, this may lead to a process framed pejoratively as "the trivialization of the phrase in the social sciences" (Fassin 2009) where an entity becomes a symbol of unthinking invocation rather than a concept of clear intellectual substance (Carrier 2018). For example, to see if the re-appellation of *terroir* as *tūrangawaewae* extends beyond wine growing to become semiotic for *locally grown* in Aotearoa New Zealand is an idea worth developing in the future. It is crucial to understand how each culture appropriates nature. Simultaneously, the finite niche markets of such nature pose economic vulnerabilities (Watts et al. 2005) since questions of appropriation, assimilation, dilution, and ignorance of the epistemologies have a bearing on their use. Māori knowledge and worldview mediate the relation between the *tangata* (people) and the *whenua* (land) of Aotearoa. Until agencies fulfilling functions of international development and funding as well as the state recognize the key connections between culture, production (indigenous or capitalistic) and nature, most of the influential propositions of the development apparatus with respect to provenance or ethics will only have short-term effects (Gonzales 2000).

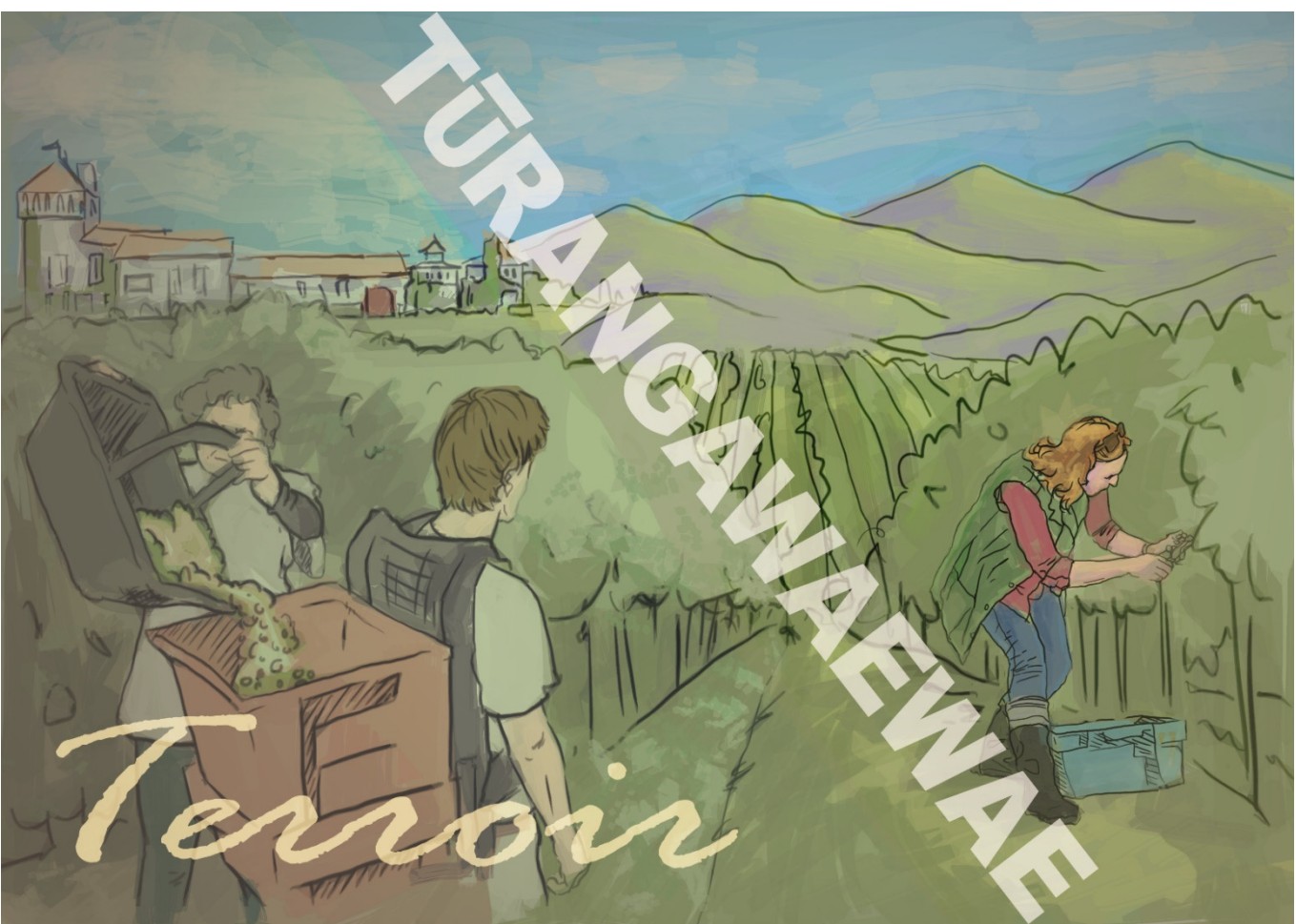

**Figure 1.** Conceptualization of two epistemologies in the context of *place*.

## 4. Producers' Role in Provenance

The non-indigenous meaning of provenance (see Section 2.2) is problematic as it favors the physical and psychological estrangement of both producers and consumers from the ecological consequences of their actions (Campbell 2009). Understanding that provenance is an outcome of dialectical tension between globalization and localization (Reid and Rout 2016), positing the *provenance* as a market niche within conventional systems to seek a premium on this would fundamentally violate its purpose of origin. By understanding the heritage of production constructed on a base of local knowledge system developed over time to make use of, or otherwise protect, local resources resists the visualization of producers as "unscrupulous" productivists. Based on Rosin's (2014) commentary of engaging the productivist ideology through utopian politics, a "new spirit of capitalism", the so-called "humanitarian" capitalism, can be initiated to prevent producers from being subsumed in the cost–benefit aspect of the market (Chakravartty and Sarkar 2013; Rosin 2014). Though it seems that a distinction can be drawn here between old capitalism, characterized by exploitation, and new capitalism, characterized by morality, but the legitimacy of the latter can also be used to challenge its common good benefits.

Developing the idea of producers' role in provenance, first and foremost an understanding of the specificities of place and the distance between universal ethics and moral experiences of local producers is critical. Secondly, regulations should occur through the realm of producers and not the other way around (Moberg 2014). Thirdly, because provenance is a shared property of supply chain actors, the content of the agreement between different parties should be managed collectively (Torre 2006). Median farm size,

infrastructure assessment, education, history, and moral experiences should be taken into consideration before presuming geography as an experimental unit. An absence of a sense of community and exclusion whereas a strong sense of collective engagement and cooperation demonstrates two styles of organizing bodies working in two different geographies (Bowen 2011; McEwan et al. 2017). Bowen (2011) mentioned that the structure of the collective organization and the relationship between supply chain actors affects how the GI, or provenance evolves and how would be its relationship with the place. A strict set of rules, for instance, 1.3 livestock units and 4600 L per hectare in the case of Comté cheese of France, helps prevent industrialization and protect artisanal methods. Considering the growing interest of big corporates, alternative forms of *economic transactions*, such as barter, donation, gifting, collecting; *working practices*, such as volunteer work, equal pay for all employees regardless of rank; economic *organization*, such as cooperatives, collectives; and *finance*, such as member loans, cooperative shares, and crowdfunding should be considered. Focusing on an understanding of *alternative*, those kinds of non-capitalist, pseudo-capitalist, or even anti-capitalist practices show the diversity of economic forms that prevent producers from direct market shock. For instance, *Ökonauten eG*, a registered cooperative in Berlin collectively purchases land through redeemable non-interest-bearing membership shares and lease it long-term to farmers that reflects production capacity, not market value (Rosol 2020). By providing long-term and secure land access to small organic farmers, *Ökonauten* pursues economic, environmental, and social goals without the pressure of fluctuating rent levels, land prices, and corporate decision-making. Traditional distribution channels should be trust- and personal interaction-based, operating through small chains, geared toward economic feasibility, and by which producers can retain a larger part of the *value* created. Focusing on inequalities, McEwan et al. (2017) demonstrated how the wealthy enjoyed inclusion and self-sufficiency in the context of cooperatives whereas others felt left and dependent. Uneven participation and prevalence of social conservatism appeared to widen rather than narrow social cleavages (Burke 2010). Informed by the fact that all places have local institutions that shape the specific moral experiences of those sharing the space, tying off all those involved in production with a single thread would stymie both cooperation and effectiveness of provenance. It is critical to understand what is at a stake, for instance, if involved producers can continue "being agriculturists". Producers and consumers, both need to keep evolving to make them distinct from the now conventional practices of the market.

Western producers have the opportunity to develop bicultural relationships with and learn from indigenous producers if they wish to change the provenance paradigm. The domestic principles of justification theory[13] (Boltanski and Thévenot 2006) gave a *feeling of community* to the producers, and this was reported as the main reason for producers to join, and they perceived each other as their extended family members (Giorda 2018). Embracing wider world views than the prevalent hegemonies are proven to appeal to consumers and therefore allow market development. The compromise between the domestic and marketing principles was suggested as a working mechanism for alternative food producers, where actors involved are looking for a *common good* (Kirwan 2006), which guides the narratives of quality used by these producers, and leads to the increased perception of product authenticity (Giorda 2018). Engaging in such non-capitalist practices, however, may be available only to those who already enjoy a certain amount of privilege (Rosol 2020). The naturalization of the alternative by big corporates makes it a further distinct and premium category, making it unreachable for marginalized populations. In the same vein, if producers themselves promote a socially inclusive market in making moral or ethical claims, there will be fewer problems in establishing provenance.

## 5. The Need for an Indigenous Construct in Provenance

Modernism, at its core, turns nature into an object that is inert—it is simply a matter without agency (Reid and Rout 2016). The fundamental clash between its aims and modernist worldview is a major issue for food provenance since modernism favors capitalism



and capitalism, in turn, favors physical abstraction (Reid and Rout 2016). This abstraction can be more concerned with the tools of certification than the moral values of fairness associated with producer participation and empowerment (McEwan et al. 2017). In-between the pursuit of economic development and care for people and ecosystems, the distinction between *alternative* and *conventional* is gradually misaligning. Simultaneously, conventional producers and retailers have appropriated provenance, at least in non-indigenous meaning, local, organic, and Fair Trade (McEwan et al. 2017; Rosol 2020). Some of those instances have been discussed where corporates are providing local, organic food and using certifications as a symbol rather than a concept. This leads to the proliferation of what becomes class-based diets such as Keto, Atkins, Weight Watchers, and similar, and a widening gap between privileged and marginalized consumers. Such massive conventionalization is a testimony to the limits of individualized consumerist framings, *quality*, and *defensive localism* via certifications, which neglect social and economic conditions of production and consumption. Mostly, universal notions of fairness do not align with local concerns, such as Fairtrade initiatives (McEwan et al. 2017; Moberg 2014). For the same reason, Watts et al. (2005) emphasized *network*—SFSCs along with *quality* aspects to stop *alternative* circulation in conventional chains and rendering their alternativeness questionable. Unfortunately, the conventionalization of localization or environmental degradation is actually expanding in the consumer world, which will ultimately bolster capitalism. The innate nature of capitalism favors exploitation and slavery (Marx 1990). Others also echoed this danger of subordination (Reid and Rout 2016; Rosol 2020) and Rosol (2020) added the third pillar, so-called *alternative economies* in an attempt to make *alternative* systems more multifaceted.

For indigenous producers, the relationship with the land as "First Nations", "Aboriginal", "First Peoples", or "Tangata whenua" (lit. "People of the land" in the Māori language), defines them, not as food producers, but as people operating within a relationship with the soil which stretches back over centuries or even millennia. Theirs is a relationship informed by ancient tribal wisdom, indigenous science (often communicated through oral tradition), relationships with the gods influencing the animist imperatives, and an innate sense of "belonging" defined by their long history and attachment through ancestral relationships and heritage. To be an indigenous producer or entrepreneur is to be the inheritor of tradition and lore, a modern manifestation of ancient (and often misunderstood or mischaracterized as "savage" or "barbarous") ways and an unsteady, ill-defined bridge between modern and ancient ways of doing and thinking. An indigenous producer must somehow live in two worlds, later discussed in more detail: their own worldview, remaining cognizant of heritage, tradition, ancient teachings, and the cultural imperative not to leave their peoples behind; and the modern world which demands provenance, sustainability, uniqueness (Barr et al. 2018), labels, food safety, gene patenting, use of chemical fertilizers and pesticides, and consistent supply levels and packaging. An indigenous approach could provide an alternative means of provenance food (Reid and Rout 2016), such as *mahinga kai* (the traditional Ngāi Tahu term referring to foods from the *tākiwa* (tribal area), of Ngāi Tahu *Ahikā Kai* business model, which was envisioned to protect and promote traditional relationships with *mahinga kai* by supporting sustainable commercial development (Barr et al. 2018). The *Ahikā Kai* system was developed to provide branding, accreditation, and traceability solutions for the sale of *mahinga kai*. Food marketers, in a contemporary approach, are selling provenance as the representations of people and places, which disparages from an animist approach that seeks to connect consumers into human and non-human networks of personal relationships (Reid and Rout 2016), and making Anthropocene less anthropocentric (Helne and Hirvilammi 2015). In other words, it is a case of image versus substance (Reid and Rout 2016).

Taking Māori knowledge as an example of business it would be appropriate to understand how culture is related to nature and commerce. Māori as *mana whenua* (people responsible for a particular region linked to place/*marae* through their ancestors) is impelled by *tikanga* (cultural values) to exercise *kaitiakitanga* (the exercise of guardianship) over land. They are known for maintaining sustainable food supplies by using the *whanau* or

group approach as well as their accumulated mātauranga Māori (science and cosmologies) for growing, procuring, cooking, and eating *kai* (foods) under a cultural imperative defined by relations with the *atua* (realm of the gods). From the critically important foundational sociocultural concept of *whakawhanaungatanga* (establishing and growing relationships) emerges the imperative of *manaakitanga* (sharing), which ensures that food is available for all people in the community (Wham et al. 2012). Other iwi and hapū were also familiar with the land, river, or sea and forest protection to safeguard essential species, maintain sustainable land use and promote biodiversity. Māori also maintains a relationship with *whenua* (land) that takes a whole-landscape, holistic approach known as *Ki Uta ki Tai* (from the mountains to the sea), to ensure that the focus is on the world we all live in, not merely the land we tend. Ngāi Tahu, the largest South Island Māori tribe who developed the online project, *Ahikā Kai*, aimed to connect *Ahikā* food producers with consumers. Keeping the animist worldview as the central point of the project, five fundamental principles formed the core of this initiative, including health (*hauora*), sustainable management (*kaitiakitanga*), fairness (*whanungatanga*), care (*kaikōkiritanga*) and cultural–ecological wisdom (*tikanga*) ( Reid and Rout 2016). The indigenous worldview reflected in their culture, values and beliefs also influenced their business practices (Mrabure 2019), in which, the *Ahikā Kai* business principles were reflected (Reid and Rout 2016). Producers who wish to be accepted under this initiative can opt for a certification system in place that prompts them to "abide by best-practices". Similarly, Mrabure (2019) discussed the five cultural values inherent to Māori entrepreneurship, *whānaungatanga* (sharing, co-operation and relationship), *wairua* (spirit), *manaakitanga* (ethics of power or hospitality, kindness), *kaitiakitanga* (guardianship of environment) and *kotahitanga* (togetherness, unity). The use of the "third space", in-between or hybrid analogy between a colonizer and colonized (Bhabha 1996), creates an alternative view, where indigenous entrepreneurs are positioning themselves (Mrabure 2019; Tretiakov et al. 2020) and distancing from existing theories that have been built on colonial worldviews (Frenkel and Shenhav 2006). The third space is a post-colonial sociolinguistic *theory of identity and community*, attributed to Homi K Bhabha, which explains the uniqueness of each person, actor or context as a "hybrid". His idea of hybridity is highly relevant as it is not only double-voiced but is also double-language, with two individual consciousness comes together and consciously fight (Barr et al. 2018); it is an attempt to construct cultural authority within conditions of political antagonism or inequality, where it continually transforms itself according to the dynamics of cultural interaction. Wayuu and Māori entrepreneurs of Colombia and Aotearoa New Zealand ostensibly use this so called third space, or other similar concepts such as n-Culturals (Pekerti and Thomas 2016), when operating in the mainstream culture of business environment. Pekerti and Thomas ( 2016) described the concept of n-Culturals where actors are able to apply more than one set of cultural values concurrently, without frame switching. Barr et al. (2018) echoed about similar ontological third space where actors perform balancing and contradicting acts.

Departing from the abovementioned colonial "production-oriented" measurement model, Māori entrepreneurs with culturally constituted orientation were found defining success by having *whanau* (family) and *hapori whānui* (wider community) wellbeing and participation. The emphasis remains on not merely growing food but disseminating the ancient tikanga of science, knowledge, and wisdom that enables more to reconnect with cultural roots to allow them to duplicate and extend the process. The foundations for wellbeing come through *kaitiakitanga* (stewardship of all our resources), *manaakitanga* (care for others), *ōhanga* (mutual prosperity) and *whanaungatanga* (connections between us), which subsequently support the development of four capital stocks: financial and physical capital, human capital, social capital and natural capital (Wolfgramm et al. 2020). Thanks to the collectivist values inherent in indigenous cultures, indigenous businesses are typically family or tribal businesses, where a kinship network between immediate family and local Indigenous community is involved in running the business (Tretiakov et al. 2020). This differs greatly from the Western entrepreneurial orientation, which is based on profit-making linked with an individualistic and neoliberal economy (Tretiakov

et al. 2020) and anonymous-but-controlling shareholding governance. The collectivist framework promotes collective leadership, negotiation and cooperation, reciprocal sharing of resources, intergenerational learning, care, and responsibility (Dudgeon and Bray 2019), as well as the free exchange of knowledge between growers. Ngāi Tahu's *Ahikā Kai* indigenous business model of encouraging entrepreneurial eco-systems of firms rather than independent individual firms assures the benefits of collectivism, such as bring potential competitors together under horizontal alliance and promote Māori *tino rangatiratanga* (self-determination) and *mana motuhake* (proudly self-governing) (Barr et al. 2018). *Ahikā Kai* provides a web-based platform for online ordering, by which producers practice and maintain *rangatiratanga*, a traceability function by which consumers connect with the accreditation system, where Ngāi Tahu assures authenticity and provenance, and a communication function, by which key principles and animism are reinforced. It was also observed that indigenous entrepreneurs are increasingly unified in their perspective of worldview, values, beliefs, and identity to construct the new meanings of business success. This leads to intentionally positioning themselves in the third or hybrid spaces. This assumption of hybridity supports the fact that social–cultural contexts matter to the meaning of success in an indigenous context. Globally, gardens are evolving as a key response of indigenous people to a food crisis (Rudolph and McLachlan 2013; Stein et al. 2018) as well as a redemocratising of land ownership and control.

The pre-colonial communal social structure brought cohesiveness in Māori land relationships. Māori always saw working the land for which they were responsible (the exercise of their identity as *mana whenua* noted above) as the exercise of *mana motuhake* (self-determination) and, even more importantly in the light of the Treaty of Waitangi, a declaration of *Tino Rangatiratanga* or absolute sovereignty over their lands. Importantly, *tino rangatiratanga* is an expression of communal ownership within the tribe or subtribe, collective responsibility for sustainability, a determination for decision-making to be inclusive and participatory, but without control of colonizing or governmental authority. It also allowed for the subversion of colonial power constructs, such as the ignorance of the government about the place of mana wahine (women) in Māori society, and the power dynamics of tohunga (specialist practitioners). It was this collectivist system of ownership and land management that was an anathema to the early colonial government, which Chief Land Purchase Commissioner Donald McLean, derided as "their present system of communism" and sought to dissolve in the 1850s (Riseborough and Hutton 1997), ultimately leaving Māori alienated from most of their ancestral lands, despite the egregious breach of the provisions of Article 2 of the Treaty of Waitangi which this policy represents. This conceptually and epistemologically separates them from the neoliberal, self-actualization, and individualistic meaning of success in entrepreneurial concepts of today (Mrabure 2019).

Among other disparities, health disparities also remain a big concern, despite globally coordinated intervention programs by various national or international agencies, among marginalized populations. The lack of access to healthy and nutritious (Stein et al. 2018) or special foods could be due to many reasons, such as unavailability, detachment, or severance from traditional land or resources (Wham et al. 2012). The inherent lack of democracy in the food system is a tangible reason for food poverty as food scarcity in itself is not a problem (Stein et al. 2018). This detachment emphasizes the importance of *whānaungatanga* as a value, or *whakawhānangatanga* (the construction of relationships) as a process, where all individuals are collectivized into relationships and encouraged to support each other as a unit (Wham et al. 2012). An additional difference in the worldview is the level of spirituality among the indigenous groups especially as it forms part of their business practices (Mrabure 2019). Specifically, a Māori worldview consists of natural, social and spiritual worlds and the connectivity between them makes this worldview holistic (Mrabure 2019). The communal view of humanity (ngā tangata katoa), land (*whenua*) and sea (*moana*) through cultural practices, such as *powhiri* (ceremony of encounter) made what *mātauranga Māori* (Māori knowledge) has developed into, built over many

generations and this is what Māori culture offers the world as a nexus of learning to improve health and nutritional wellbeing in current food system models (Wham et al. 2012). Historic place-based or context-based indigenous knowledge systems, values, and lifeways (Turner et al. 2013) should be seen as a model for future developmental projects so as not to replicate the mistakes of the past.

The animistic concept of *mauri* (spirit) guides a considerable amount of Māori engagement with the land. It is variously described as "energy which binds and animates all things in the physical world" (Royal 2007), "a vitalizing principle pertaining to things animate and inanimate" (Best 1982), and "*e kaitiaki te ora o te pae o Papatuanuku, te ora o nga hua me nga rakau katoa o Tane, otira, me te ora o te tinana o nga tangata katoa*" (healthy soils, healthy plants, healthy people) (Harmsworth 2018). Mauri remains critical for Māori to guide land and resource use. If *kaitiakitanga* is the guiding principle to restore balance into ecosystems, then the desired outcome is to (at least) maintain or ideally restore mauri, for people as well as the natural and spiritual worlds. Through adherence to the precepts of *tikanga* and the traditions of *kaitiakitanga* in planting, growing, harvesting and marketing produce, the mauri of the land, the ecosystem, the socio-cultural relationships, and the people are maintained. The use of this methodology is very old; communicating this to end-use consumers is relatively new. In the process, *Ahikā Kai* may be connecting consumers to the producers, however, the success of this initiative at large may serve as an *alternative*, more sustainable model to study and implement.

## 6. Conclusions

Provenance is a multifaceted concept that needs support from multispecies ethnography to infuse *telos*, *terroir*, social and cultural dimensions in nonhuman and human life. The concept of localness should not be confined to geographic or spatial limits but instead forged through native ecosystems, seasons, animals, people, language, rituals, and beliefs to support the non-tangible environmental, social, and emotional benefits of provenance. Understanding the growing lucrative nature of provenance, producers should complement the *quality* with short food supply chains and diversity of non-capitalist-based *economic* practices to give alterity a meaning of multidimensionality. Realizing the fact that even if direct effects may be limited, it holds the potential to be a catalyst for broader societal impact and change through transforming economic and societal relations. An effective communication strategy of the socio-cultural and spiritual intangibles to consumers should be central in traceability measures. It is critical to extract provenance from Western geopolitical and economic interests, and embed it within indigenous cultural and environmental contexts that include cosmogonic strategies and cognitive models depending upon the immediate local. Indeed, issues related to provenance need to be rephrased to identify which modern tools may be of help to indigenous and local communities rather than finding ways of integrating indigenous knowledge in the western paradigm. For example, *tūrangawaewae* instead of *terroir* would be more appropriate in the immediate local context of Aotearoa New Zealand, but simultaneously it should not be embedded in a foreign rather than the immediate Māori cosmovision.

*In toto*, a contrast between the indigenous meaning of provenance in foods and the Western view of production was presented in this review paper. These concepts do not challenge transferability but set the baseline for comparing the two different systems. In the conclusion of this review, we proposed a holistic approach to understand both ideologies but with an emphasis on Maori culture and its perspectives. This can open the room for more research work to align these parallel words that can help the producer, consumers and the environment.

At the micro-level, consumers are exercising opportunities to purchase local, with fewer food miles, based on more sustainable production, but why they are operating in such a way based on which concerns should be the question for future endeavors. Can these concerns be viewed at meso- and macro-level and if yes, what are they? Does wellbeing and communion or rematriation have any role? It should also be considered whether



integrating an existential school of thought with social justice concerns would yield better outputs. By considering the complexities of ethical consumption in a transnational and neoliberal context, would it ever be possible to draw a clear distinction between ethical and unethical practice? Simultaneously, a distinction is required to counter conventional market systems, either through alternative chains or through economics. The Māori worldview is presented as a case for an alternative framework. The Māori worldview and business perspectives are based on five fundamental principles [health (hauora), sustainable management (kaitiakitanga), fairness (whanungatanga), care (kaikōkiritanga), and cultural–ecological wisdom (tikanga)] which we suggest could be a model for sustainable and ethical food production. This Māori worldview consists of natural, social and spiritual worlds, and the connectivity between them makes this worldview holistic. Food and fiber industries around the world can learn from ancestral knowledge to improve the relationships between consumers and food products. Differences in food production systems, spatial context (ki uta ki tai), temporal context (I nga wao Anthropocene), human context (ko au te whenua, ko te whenua ko au—I am the land and the land is me), and the natural world (te Taiao) need to be taken into consideration when discussing food provenance. In a time when the local Aotearoa New Zealand government is emphasizing sustainable food and fiber production under Te Taiao initiative, understanding the provenance construct and Māori entrepreneurship framework would be of immense importance to guide researchers and policymakers on the implications and concerns.

**Author Contributions:** Conceptualization, C.S., L.C., R.H. and D.D.T.; writing—original draft preparation, C.S.; writing—review and editing, C.S., D.D.T., L.C. and R.H.; supervision, D.D.T., L.C. and R.H.; project administration, R.H. and D.D.T.; funding acquisition, R.H. and D.D.T. All authors have read and agreed to the published version of the manuscript.

**Funding:** This research was funded by the Lincoln University, Aotearoa New Zealand, through the Center of Excellence—Food for Future Consumers.

**Acknowledgments:** First and foremost, the authors would like to thank Kestin Stewart of Lincoln University for providing an exquisite figure of two cosmologically different concepts. Secondly, the authors would like to thank two anonymous reviewers for their insightful and constructive feedback on early versions of this review article. Responsibility for the rest or any of the errors is our own.

**Conflicts of Interest:** The authors declare they have no conflict of interest concerning this manuscript and its research.

## Notes

1. Adjectives were italicized to prevent confusion as Watts et al. (2005) suggests that they are easy to confuse with their counterparts in economics.
2. Not regulated by social institutions other than the *market*.
3. From Polanyi's dichotomous view of economic systems: *Household/Reciprocity/Redistribution* versus *Market*.
4. Economic externalities are, by definition, the costs incurred or the benefits received by third parties following economic production or consumption.
5. Intentionally avoiding the use of word 'moral' here because economy itself is more or less moral.
6. A tribe, inhabiting Garam Basti, under Kalchini Block in Alipurduar district of West Bengal, India.
7. It is a process whereby societies come under or adopt Western culture in areas such as industry, economics, lifestyle, customs, etc. Some critiques assume westernization to be equivalent of modernization.
8. which is not absorbed in their origins.
9. By in-place author(s) means knowledge systems, symbols or behaviors rooted in a specific territorial context, used to evoke certain values that develop and cultivate the aesthetic dispositions of alternative, such as "family-owned", "homemade", craftsmanship, etc.
10. For the same reasons, they have been *neutralized* to subordinate within the conventional market system via *quality* and *defensive localism* (Watts et al. 2005).
11. Territory is defined by José Muchnik (2010) as a space that is 'socially constructed, culturally marked, and institionally regulated'.

[12] The marketplace is 'both a specific geographical place and a localized set of social institutions, transactions, social actors, organzations, products, trade practices, and cultural meanings motivated by a wide variety of factors including, but not limited to, "purely economic" or "market" forces' (Bestor 2004).

[13] Boltanski and Thévenot (2006) create a framework to evaluate judgements of worth within six different worlds of values, namely, (i) the inspired world, (ii) the domestic world, (iii) the world of fame, (iv) the civic world, (v) the market world, (vi) and the industrial world.

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
