# Peer review of "Indigenous Meanings of Provenance in the Context of Alternative Food Movements and Supply-Chain Traceability: A Review"

_socsci, doi:10.3390/socsci10070255_

Round 1

Reviewer 1 Report

The paper is of an review nature. Many literature items have been used, however, it has also revealed some imperfections and needs to be corrected in order to be accepted. First of all, the aim of the presented study was not precisely stated. Broad and sometimes quite complicated considerations presented in the paper may be incomprehensible to the reader, because it is not clear what is the aim of the study and what the authors want to prove. The paper has the value resulting from the literature review, but this is not enough to prove its scientific or utilitarian value. The discussed topic could be of interest to scientists and practitioners dealing with the problems of food, food trade and agricultural policy, if the authors set a specific goal and complete the paper with specific conclusions. In my opinion, disputing the spatial confinement practices of stakeholders and governments that are currently applied to the concepts of food provenance has not been duly proven. In particular there is no empirical material in the paper.

Author Response

Thanks for the generalized assessment of this work. This research uniquely engages with te ao Māori/the Māori world in reconceptualizing provenance and examining place as a unique positionality for producers. In taking this innovative approach, this paper sets the kaupapa/basis for future work examining the nexus of indigenous relationships with land and markets. An empirical approach, if taken before consideration of the contributing literature, detail of epistemologies, and how indigeneity may form and inform research would drive consideration of provenance into an early, narrow intellectual range of cul-de-sacs. We expect future research to springboard off this study and we welcome future developments. Overall, the unique approach in bringing indigenous animist spirituality into provenance is of sufficient moment to resist specifics as requested. Moreover, to improve the utilitarian value of the paper, some sections have been rearranged and modified throughout concerning the aim of the paper. We also added some more insights to strengthen the incompetency of certifications, such as from McEwan et al. (2017), Orzes (2017), Flachs (2017), Moberg (2014), and Kumbamu (2018).

Reviewer 2 Report

Thanks for submitting an interesting manuscript! Here a general comment:
I feel the paper pays too much attention on defining attributes of and criticising the Western conceptions in alternative food systems and supply chains. Emphasis could instead be laid on revealing the qualities of the provenance or place conceptions as based on indigenous insight. That means their holistic quality which could be applied to and add value to supply chains. However, I don’t see this argument realised consistently in this review.
Another feeling that emerged while reading the paper is overloaded of concepts and definitions and references to world views and economic systems, while these do not necessarily help reveal the advantages of considering indigenous provenance notions. You could take away a lot of the paper’s heaviness by putting Section 6 more prominently and extensively.

Another aspect that does not reveal to me is the differentiation between place and provenance. I feel that the argumentation does not benefit from differentiation between both. Why is ‘place’ from a Maori point of view not sufficient in order to suggest a holistic view on product-place relations in food system? What is the additional benefit in introducing ‘provenance’?

I believe the paper would benefit from restructuring and reflecting on the key arguments. Any way, I find the content valuable and very relevant to social sciences research, and policy, and I would love to see it published!

Below my specific comments:

L32: local issues – find other term

L48-55: Please provide a reference for this Is that from your own empirical insight?

L59: I understand your argument and agree: Already from a linguistic perspective, the semantic of ‘provenance’ is beyond spatial. In my eyes, this however applies to ‘origin’ too. It contains a socio-cultural dimension. I therefore suggest to either replace ‘origin’ by ‘place’ (although place also has social meaning and can be seen as socially constructed), or add some attribute such as ‘spatial’ or ‘space-related’ to ‘origin’.

L63: I think your reference to ‘modernity’ needs to be explained briefly, 1-2 lines will be sufficient.

L68 is a core sentence for me within this Introdcution. I suggest putting it more prominently, e.g. in Line 63 just before ‘. Provenance’.

L73-75: I support your agenda of considering socio-cultural and economic spheres in the concept of provenance. However, I find the wording quite absolute: I’d say provenance is workable and deliverable in isolation, but considering other beyond spatial spheres provides more and more qualitative understanding of food systems, its involved stakeholders and perhaps human-nature relationship. I suggest you may strengthen this argument by taking the absoluteness from ‘it cannot be considered in isolation’ or ‘is not deliverable’.

L79: I’d like to see a your agenda with and structure of this paper. E.g., Section 2, the Western paradigm is not at all mentionned in the introduction.

L102-104: Please mention source for the first time already at the end of this sentence.

Section 2.: I did not see clearly how the cited literature and concepts represent the Western modernist paradigm, and how they relate to the concept of provenance. I suggest to discuss a greater diversity of literature without stretching the section. After having read the following sections, I would also tend to rename the heading. The Western paradigm should not be the main title of the section but a subtitle equal in weight to 2.1 and 2.2.

L173: I wonder, aren’t the economic externalities corresponding to the production of material wealth? What do you mean by ecomonic externalities? Maybe this question is due to my non-economical background.

L214-216: Great! Here you relate back to Section 2.

L293: New paragraph.

L303-308: Great and important argument! To support your overall argumentation of your paper, I suggest to move these lines up into Section 1.

L332-337: For better structure, move this up, before introducing concepts of animism from different regions.

L356: Bring together with previous mention on deep ecology in L332.

L374: Use term different to ‘illogical’ to defend and strengthen your valuable argument. Different ontologies produce different logics of provenance, I’d say.

Section 2.1: I feel that content up to L260 would be better placed under and discussed with content of Section 2, modernist / Western paradigms. You start discussing indigeneous views in L261, hence make a section here. I further see potential to condense and shorten your argument, which could be done by merging L167-260 with 2.

L462: ‘talking something’ – add ‘about’?

Section 2.2 is an interesting section, which, I feel, should be merged with 2. The different schools of thought that you introduce in L439 come a bit abruptly and seem detached from the rest of the section. The argument would benefit from relating them to the preceding paragraphs, since they are an important derivation of modern paradigms.

Table 1: Layout: (Opposing) line should be aligned. Some references appear for the first time in the table. Shouldn’t they be explained in the text before? Where in text do you refer to Table 1? It should not stand alone but in relation to the text. It is not self-explaining. I therefore wonder whether it is needed at all.

472: Is this your insight? Otherwise cite.

L517: You might want to mention the ecosystem thinking, also ecosystem services. Biodiversity is often used in that context, as a desired asset of the ecosystem. Ecosystem itself reflects the dualistic view (nature caters for the human needs).

L529-531: This is very central to your argument in my eyes.

L533: Add ‘cf. Section 2.2’

L534: Foot note is not self-explaining I think.

L540: ‘may be’ instead of ‘maybe’?

L549: New paragraph before ‘Food provenance’.

L577-578: Structure of sentence appears confusing..

L606: New paragraph before ‘SFSCs have?

L615ff.: I understand the point of alternative food systems being rather exclusive. To bring in my own experience, I have worked on organic food movements e.g. in Thailand and India and found various kinds of farmers markets, aligning with parallel philosophies: The ones aiming at mostly urban middle classes and lifestyle shoppers, and the lower end ones that aim at making healthy foods accessible to all. The latter almost sell at the price of the conventional market. They can do so because there’s no certification and no middle men involved. Also the CSA programmes are relatively low cost and quite popular also in semi-rural areas. What I want to say, I think this would be a good spot to bring in some literature on alternative / local food systems from non-Western spheres. They might enrich your review, and even more so given that you argue for non-Western paradigms of provenance in the food context. The literature you are citing instead mirrors mostly Western food settings.

L630: Really? Does provenance automatically restore relationship to nature?

L629-633: This argument would also be valid when ‘provenance’ is replaced by ‘place’. Consumers’ awareness of mere place of production would also turn the purchased food from nowhere into from somewhere, one could argue. I see and support your point of this qualitative understanding of provenance compared to place. I therefore recommend to reveal it more clearly within this Section 2.1. Strengthen your argument. Stress e.g. L534ff. The literature cited goes too much into local food system debates, farmers market, SFSCs. I think the potential of indigeneous approaches to provenance is understated in this section, use your cited literature only to support it. In my eyes, the advantage of referring to provenance rather than place is not salient.

L634: This is the first time you speak of ‘capitalism’. I would add a reference to later sections that use the term frequently. Or explain here, how you relate capitalism and provenance, maybe through the fetishism, e.g.

L653: I actually find the concept of terroir quite rich and relatable, not abstract…

L658: What is the reason for bringing in climate change here?

L726-729: Great! A pity that you can’t expand on that.

L739-744: Split sentence.

L745: The meaning of the figure does not reveal to me. In which context is it employed?

L747: ‘Producers’ role’? ‘ missing.

L824: Explain justification theory.

L801-834: How to bring these insights and provenance together ? The link is not clear enough.

L835ff.: This seems like a recommendation. I would move that into conclusions.

Section 4.: The role of producer in co-producing provenance remains disclosed to me to be honest. Also, if we think in terms of Western producers, how can they help change the provenance paradigm if they adhere to the same prevalent ontologies (of provenance, of being in general) and world views?

L860: better understanding?

L878: Indeed, I experienced the same.

L911: Remove first full stop.

L917-920: I would avoid these generalisations. I don’t think they are valid.

L929: The paper would benefit from citing of other authors. Rosol 2020 appears quite often considering that your intention is a review.

L940: This gets onto the track of consumer behaviour while the link to provenance gets lost.I see potential to shorten this section here.

L951f.: repetition

Section 5: What is the link to the indigeneous concepts of provenance here?

L967ff.: To be honest, I do not see where the analysis suggests that…

L990: The following part supports your argument.

L1131: Very valuable aspect!

L1147: It is old indeed, and similar concepts probably exits around the globe. Maybe mention that the Maori conception is just one example?

Section 6 is the key section for me. And I think it should have far more weight in your analysis. Sections 3, 4 and 5 could be reduced significantly. Another thought, why does Section 6 stand alone? It could be merged into 3, providing a dialogue-like structure, in opposition to the non-indigenous conceptions. Or, it could be placed before Section 3.

L1166: Start new line.

L1179: But before you raise doubts about transferability of the Maori conception right?

Author Response

General comment:

I feel the paper pays too much attention on defining attributes of and criticising the Western conceptions in alternative food systems and supply chains. Emphasis could instead be laid on revealing the qualities of the provenance or place conceptions as based on indigenous insight. That means their holistic quality which could be applied to and add value to supply chains. However, I don’t see this argument realised consistently in this review. Another feeling that emerged while reading the paper is overloaded of concepts and definitions and references to world views and economic systems, while these do not necessarily help reveal the advantages of considering indigenous provenance notions. You could take away a lot of the paper’s heaviness by putting Section 6 more prominently and extensively. Another aspect that does not reveal to me is the differentiation between place and provenance. I feel that the argumentation does not benefit from differentiation between both. Why is ‘place’ from a Maori point of view not sufficient in order to suggest a holistic view on product-place relations in food system? What is the additional benefit in introducing ‘provenance’?

Thanks you for the overall assessment of the work. We tried our best to accommodate as many changes as we can to make it less bulky and relevant to the concept of interest. In the process, we changed the structure of the paper and also removed some sections to make it more clear and coherent. We consciously introduced economic systems because they are often get neglected in sociology literature despite their crucial role in society. We also removed section 5 – consumers’ perceptions about provenance to reduce the paper’s heaviness and shortened other sections too. We believed that these changes have improved the clarity and conciseness of this paper.

I believe the paper would benefit from restructuring and reflecting on the key arguments. Anyway, I find the content valuable and very relevant to social sciences research, and policy, and I would love to see it published!

Below my specific comments:

L32: local issues – find other term

This change has been made in the revised manuscript.

L48-55: Please provide a reference for this Is that from your own empirical insight?

It is from our own findings. We recently finished a focus group study where south Asian (Chinese) participants evoke the safety concerns of China-originated milk and milk products.

L59: I understand your argument and agree: Already from a linguistic perspective, the semantic of ‘provenance’ is beyond spatial. In my eyes, this however applies to ‘origin’ too. It contains a sociocultural dimension. I therefore suggest to either replace ‘origin’ by ‘place’ (although place also has social meaning and can be seen as socially constructed), or add some attribute such as ‘spatial’ or ‘space-related’ to ‘origin’.

Thanks for the suggestion. This change has been made.

L63: I think your reference to ‘modernity’ needs to be explained briefly, 1-2 lines will be sufficient.

Thanks for the advice. This change has been made.

L68 is a core sentence for me within this Introdcution. I suggest putting it more prominently, e.g. in Line 63 just before ‘. Provenance’.

Thanks for the suggestion. This change has been made.

L73-75: I support your agenda of considering socio-cultural and economic spheres in the concept of provenance. However, I find the wording quite absolute: I’d say provenance is workable and deliverable in isolation, but considering other beyond spatial spheres provides more and more qualitative understanding of food systems, its involved stakeholders and perhaps human-nature relationship. I suggest you may strengthen this argument by taking the absoluteness from ‘it cannot be considered in isolation’ or ‘is not deliverable’.

Thanks for the note. This change has been made to accommodate the suggestion.

L79: I’d like to see a your agenda with and structure of this paper. E.g., Section 2, the Western paradigm is not at all mentionned in the introduction.

Thanks for the suggestion. The agenda has been added.

L102-104: Please mention source for the first time already at the end of this sentence.

Thanks, the source has been added.

Section 2.: I did not see clearly how the cited literature and concepts represent the Western modernist paradigm, and how they relate to the concept of provenance. I suggest to discuss a greater diversity of literature without stretching the section. After having read the following sections, I would also tend to rename the heading. The Western paradigm should not be the main title of the section but a subtitle equal in weight to 2.1 and 2.2.

Section has been renamed “Transformation of marketplaces and consumer activism” to align it with the need of provenance.

L173: I wonder, aren’t the economic externalities corresponding to the production of material wealth? What do you mean by ecomonic externalities? Maybe this question is due to my noneconomical background.

No, economic externalities do not mean material wealth but they are the unseen costs associated with exchanges, irrespective of their nature (economical or non-economical). Economic externalities are, by definition, the costs incurred or the benefits received by third parties following economic production or consumption. The key factor about this concept is that the effect of the externalities is not included in the price of or returns to production, a situation that has led to the generation of polluter-pays charges and user-pays pricing adjusted for environmental costs.

L214-216: Great! Here you relate back to Section 2.

Thanks for the comment.

L293: New paragraph.

This was addressed in the revised manuscript.

L303-308: Great and important argument! To support your overall argumentation of your paper, I suggest to move these lines up into Section 1.

Thanks for the suggestion, this change has been made.

L332-337: For better structure, move this up, before introducing concepts of animism from different regions.

Thanks for the suggestion, this change has been made.

L356: Bring together with previous mention on deep ecology in L332.

Thanks for the suggestion. This change has been made.

L374: Use term different to ‘illogical’ to defend and strengthen your valuable argument. Different ontologies produce different logics of provenance, I’d say.

Thanks for the suggestion. This change has been made.

Section 2.1: I feel that content up to L260 would be better placed under and discussed with content of Section 2, modernist / Western paradigms. You start discussing indigeneous views in L261, hence make a section here. I further see potential to condense and shorten your argument, which could be done by merging L167-260 with 2.

Thanks for the suggestion. This change has been made.

L462: ‘talking something’ – add ‘about’?

This was added in the revised manuscript.

Section 2.2 is an interesting section, which, I feel, should be merged with 2. The different schools of thought that you introduce in L439 come a bit abruptly and seem detached from the rest of the section. The argument would benefit from relating them to the preceding paragraphs, since they are an important derivation of modern paradigms.

Thank you for the note. We added a new connecting line in-between both paragraphs. However, we retained the structure of the paper.

Table 1: Layout: (Opposing) line should be aligned. Some references appear for the first time in the table. Shouldn’t they be explained in the text before? Where in text do you refer to Table 1? It should not stand alone but in relation to the text. It is not self-explaining. I therefore wonder whether it is needed at all.

As per your suggestion, table has been removed.

472: Is this your insight? Otherwise cite.

No, it wasn’t. I added citation.

L517: You might want to mention the ecosystem thinking, also ecosystem services. Biodiversity is often used in that context, as a desired asset of the ecosystem. Ecosystem itself reflects the dualistic view (nature caters for the human needs).

Changes have been made to accommodate the suggestion.

L529-531: This is very central to your argument in my eyes.

Thanks for the note.

L533: Add ‘cf. Section 2.2’

Thanks for the note, change has been made.

L534: Foot note is not self-explaining I think.

Thanks for the note. We rephrased it to be clearer.

L540: ‘may be’ instead of ‘maybe’?

This was addressed in the revised manuscript, thanks for the not.

L549: New paragraph before ‘Food provenance’.

This was addressed in the revised manuscript.

L577-578: Structure of sentence appears confusing.

This was addressed in the revised manuscript.

L606: New paragraph before ‘SFSCs have?

This was addressed in the revised manuscript.

L615ff.: I understand the point of alternative food systems being rather exclusive. To bring in my own experience, I have worked on organic food movements e.g. in Thailand and India and found various kinds of farmers markets, aligning with parallel philosophies: The ones aiming at mostly urban middle classes and lifestyle shoppers, and the lower end ones that aim at making healthy foods accessible to all. The latter almost sell at the price of the conventional market. They can do so because there’s no certification and no middle men involved. Also the CSA programmes are relatively low cost and quite popular also in semi-rural areas. What I want to say, I think this would be a good spot to bring in some literature on alternative / local food systems from non-Western spheres. They might enrich your review, and even more so given that you argue for non-Western paradigms of provenance in the food context. The literature you are citing instead mirrors mostly Western food settings.

Thanks for the suggestion. We added more literature especially from the non-Western sphere to strengthen the concept.

L630: Really? Does provenance automatically restore relationship to nature?

Thanks for the remark. Provenance, in connection to place, restore relationships with nature.

L629-633: This argument would also be valid when ‘provenance’ is replaced by ‘place’. Consumers’ awareness of mere place of production would also turn the purchased food from nowhere into from somewhere, one could argue. I see and support your point of this qualitative understanding of provenance compared to place. I therefore recommend to reveal it more clearly within this Section 2.1. Strengthen your argument. Stress e.g. L534ff. The literature cited goes too much into local food system debates, farmers market, SFSCs. I think the potential of indigeneous approaches to provenance is understated in this section, use your cited literature only to support it. In my eyes, the advantage of referring to provenance rather than place is not salient.

Thanks for the remarks. This section was re-written for a clearer understanding.

L634: This is the first time you speak of ‘capitalism’. I would add a reference to later sections that use the term frequently. Or explain here, how you relate capitalism and provenance, maybe through the fetishism, e.g.

The mentioned line has been deleted for its non-relevance with the preceding concepts.

L653: I actually find the concept of terroir quite rich and relatable, not abstract…

Thanks for the note. The meaning of this concept has changed in the revised manuscript.

L658: What is the reason for bringing in climate change here?

We removed the particular phrase because of its irrelevance.

L726-729: Great! A pity that you can’t expand on that.

Thanks for the note.

L739-744: Split sentence.

This was addressed in the revised manuscript, thanks.

L745: The meaning of the figure does not reveal to me. In which context is it employed?

Figure was conceptualized in the context of place.

L747: ‘Producers’ role’? ‘ missing.

Thanks for the note. Mistake has been addressed.

L824: Explain justification theory.

Theory has been explained in footnote. Thanks for your comment.

L801-834: How to bring these insights and provenance together? The link is not clear enough.

We tried to make this connection clearer with rephrasing some of the content

L835ff.: This seems like a recommendation. I would move that into conclusions.

Thanks for your suggestion. This change has been made.

Section 4.: The role of producer in co-producing provenance remains disclosed to me to be honest. Also, if we think in terms of Western producers, how can they help change the provenance paradigm if they adhere to the same prevalent ontologies (of provenance, of being in general) and world views?

This changes has been made to connect the producers' role clearly.

L860: better understanding?

This change has been made. Thanks.

L878: Indeed, I experienced the same.

Thanks for sharing your experience. I came from a rural family and during my childhood, I kept asking my mom that do not use wood sticks or paper or plastic, or cow-dung cakes to make food because we are making a lot of smoke and smoke is causing pollution. I was concerned in a time when I had not many resources or economic capital, so, how I can say that poor are not concerned.

L911: Remove first full stop.

Thanks. This change has been made.

L917-920: I would avoid these generalisations. I don’t think they are valid.

Thanks for the suggestion. This change has been made.

L929: The paper would benefit from citing of other authors. Rosol 2020 appears quite often considering that your intention is a review.

Thanks for the suggestion. We made some changes to the body to accommodate more relevant references as per requirements.

L940: This gets onto the track of consumer behaviour while the link to provenance gets lost. I see potential to shorten this section here.

Thanks for the suggestion. This change has been made.

L951f.: repetition

Thanks for the note. This change has been made.

Section 5: What is the link to the indigeneous concepts of provenance here?

Thanks for the argument. We made changes to connect provenance with consumers.

L967ff.: To be honest, I do not see where the analysis suggests that…

Line has been deleted.

L990: The following part supports your argument.

Thanks.

L1131: Very valuable aspect!

Thanks.

L1147: It is old indeed, and similar concepts probably exits around the globe. Maybe mention that the Maori conception is just one example?

Thanks for the suggestion.

Section 6 is the key section for me. And I think it should have far more weight in your analysis.

Thanks for your suggestion.

Sections 3, 4 and 5 could be reduced significantly. Another thought, why does Section 6 stand alone? It could be merged into 3, providing a dialogue-like structure, in opposition to the nonindigenous conceptions. Or, it could be placed before Section 3.

As per your suggestion, sections 3 and 4 have been shortened. While section 5 has been removed to keep readers focus on provenance and not on consumers.

L1166: Start new line.

Thanks for the suggestion. This change has been made.

L1179: But before you raise doubts about transferability of the Maori conception right?

Round 2

Reviewer 1 Report

The paper has been improved and is of better quality. I do not entirely agree with the presented reasoning, but I believe that the paper may be published in its current form.

Author Response

We thank the reviewer(s) for their insightful comments that have improved the clarity and content of this review paper. We acknowledge that the subject of this paper “provenances in foods” might be exposed to multiple interpretations, and some of these ideas might contradict each other. We have discussed these concepts and definitions taking into consideration different fields of studies; however, we consider that provenance of foods is still an evolving subject that will be further discussed in future works. This work is taking the first step in challenging the traditional ideas of the provenance of foods by adding an indigenous context into the meaning.

Reviewer 2 Report

Dear authors,

I enjoyed reading your revised manuscript and believe it has improved immensely! It reads very well, and your arguments are clear and distinct. I look forward to seeing it published finally.

Below a few suggestions for minor changes.

All the best.

L18: Maybe add “we suggest that” provenance should be embedded...

L21: development projects? New…

L27: Replace “This review also” by “It”.

L61: Own findings – I’m afraid this is not sufficient. What about referring to field notes, e.g. “from field notes 2021” pr “own findings 2021”? In case reader claim the source, I guess you’ll need to provide it.

L68: Can you replace the URL by name of site? URL is for the list of references only, I thought.

L92-96: Delete the referencees?

L110/111: “through” not “thru”

Section 1: Improved lots!

L381: Indeed. Nice point.

L412: See comment on L68.

L727/774: “through”

Foot note 16: “Territory”

L1560 / 1570: Great!

Author Response

Dear authors,

I enjoyed reading your revised manuscript and believe it has improved immensly! It reads very

well, and your arguments are clear and distinct. I look forward to seeing it published finally.

The authors are thankful to you for making this happen.

Below a few suggestions for minor changes.

All the best.

L18: Maybe add “we suggest that” provenance should be embedded...

The change has been made.

L21: development projects? New…

Thanks, the change has been made.

L27: Replace “This review also” by “It”.

The change has been made.

L61: Own findings – I’m afraid this is not sufficient. What about referring to field notes, e.g. “from

field notes 2021” pr “own findings 2021”? In case reader claim the source, I guess you’ll need to

provide it.

Thanks for the suggestion. Yes, it would be better.

L68: Can you replace the URL by name of site? URL is for the list of references only, I thought.

Thanks, we replaced the URL.

L92-96: Delete the referencees?

Thanks. The change has been made.

L110/111: “through” not “thru”

Thanks for the note. The change has been made.

Section 1: Improved lots!

Thanks for the appreciation.

L381: Indeed. Nice point.

Thanks.

L412: See comment on L68.

Thanks for the note. The change has been made.

L727/774: “through”

Thanks for the note, the change has been made.

Foot note 16: “Territory”

Thanks, the typo has been corrected.

L1560 / 1570: Great!

Thanks.
